# Skin Temperature Circadian Rhythms and Dysautonomia in Myalgic Encephalomyelitis/Chronic Fatigue Syndrome: The Role of Endothelin-1 in the Vascular Tone Dysregulation

**DOI:** 10.3390/ijms24054835

**Published:** 2023-03-02

**Authors:** Trinitat Cambras, Maria Fernanda Zerón-Rugerio, Antoni Díez-Noguera, Maria Cleofé Zaragozá, Joan Carles Domingo, Ramon Sanmartin-Sentañes, Jose Alegre-Martin, Jesus Castro-Marrero

**Affiliations:** 1Department of Biochemistry and Physiology, School of Pharmacy and Food Sciences, University of Barcelona, 08028 Barcelona, Spain; 2Department of Nutrition, Food Science and Gastronomy, Food Science Torribera Campus, University of Barcelona, 08921 Barcelona, Spain; 3Nutrition and Food Safety Research Institute, INSA-UB, 08921 Barcelona, Spain; 4Clinical Research Department, Laboratorios Viñas, 08012 Barcelona, Spain; 5Department of Biochemistry and Molecular Biomedicine, Faculty of Biology, University of Barcelona, 08028 Barcelona, Spain; 6Division of Rheumatology, ME/CFS Clinical Unit, Vall d’Hebron University Hospital, Universitat Autònoma de Barcelona, 08035 Barcelona, Spain; 7Division of Rheumatology, ME/CFS Research Unit, Vall d’Hebron Research Institute, Universitat Autònoma de Barcelona, 08035 Barcelona, Spain

**Keywords:** chronic fatigue syndrome, circadian rhythms, endothelin-1, myalgic encephalomyelitis, skin temperature

## Abstract

There is accumulating evidence of autonomic dysfunction in myalgic encephalomyelitis/chronic fatigue syndrome (ME/CFS); however, little is known about its association with circadian rhythms and endothelial dysfunction. This study aimed to explore the autonomic responses through an orthostatic test and analysis of the peripheral skin temperature variations and vascular endothelium state in ME/CFS patients. Sixty-seven adult female ME/CFS patients and 48 healthy controls were enrolled. Demographic and clinical characteristics were assessed using validated self-reported outcome measures. Postural changes in blood pressure, heart rate, and wrist temperature were recorded during the orthostatic test. Actigraphy during one week was used to determine the 24-h profile of peripheral temperature and activity. Circulating endothelial biomarkers were measured as indicators of endothelial functioning. Results showed that ME/CFS patients presented higher blood pressure and heart rate values than healthy controls in the supine and standing position (*p* < 0.05 for both), and also a higher amplitude of the activity rhythm (*p* < 0.01). Circulating levels of endothelin-1 (ET-1) and vascular cell adhesion molecule-1 (VCAM-1) were significantly higher in ME/CFS (*p* < 0.05). In ME/CFS, ET-1 levels were associated with the stability of the temperature rhythm (*p* < 0.01), and also with the self-reported questionnaires (*p* < 0.001). This suggests that ME/CFS patients exhibited modifications in circadian rhythm and hemodynamic measures, which are associated with endothelial biomarkers (ET-1 and VCAM-1). Future investigation in this area is needed to assess dysautonomia and vascular tone abnormalities, which may provide potential therapeutic targets for ME/CFS.

## 1. Introduction

Myalgic encephalomyelitis, also known as chronic fatigue syndrome (ME/CFS), is a debilitating multifaceted disorder that affects more than 50 million people worldwide. ME/CFS is a chronic condition that predominantly affects women [1]. The etiology of ME/CFS remains unknown, but it appears to be multifactorial, with immunogenetic and environmental factors influencing disease onset [2]. ME/CFS is characterized by debilitating post-exertional fatigue, neurocognitive impairments, autonomic dysfunction, and nonrestorative sleep, causing a marked reduction in daily activities, especially in women [3,4]. Currently, there are no reliable diagnostic markers for ME/CFS, nor are any FDA-approved disease-modifying drugs available [5].

Previous reports on autonomic dysfunction in ME/CFS suggest an imbalance between the sympathetic and parasympathetic nervous systems, with decreased parasympathetic tone and increased sympathetic output [6]. Autonomic symptoms are highly prevalent in ME/CFS and include dizziness, orthostatic hypotension, palpitations, lipotimia, hot flashes, constipation, and night sweating [7]. Moreover, hemodynamic determinants related to blood pressure (systolic pressure, diastolic volume, cardiac output, heart rate variability, arterial stiffness, cerebral blood flow, etc.) have also been described to be disturbed in ME/CFS [8,9,10]. Dysregulation of the autonomic nervous system also leads to an impaired vasomotor response in ME/CFS, such as postural orthostatic tachycardia syndrome (POTS). However, there are some discrepancies in the prevalence of POTS, ranging from 23.7% in ME/CFS compared with only 4% in healthy controls [11] to 5.7% in ME/CFS, without differences with respect to non-ME/CFS individuals [12]. A recent review paper noted that circadian rhythm disruptions (sleep activity, insomnia, cognition problems, energy disturbances, impaired thermoregulation and dysautonomia) and cytokine profiling (mainly TGF-β) may be implicated in ME/CFS and long COVID. However, to date, no association has been established between peripheral skin temperature circadian rhythms, circulating endothelial biomarkers, and dysautonomia in ME/CFS [13]. The circadian system, whose master clock is the hypothalamic suprachiasmatic nuclei, regulates the manifestations of the circadian rhythms, including sleep schedules. The master clock regulates peripheral clocks and generates rhythms throughout the organism by means of daily variations in hormones, temperature rhythm, and regulation of the balance between the sympathetic nervous system, which peaks during the day, and the parasympathetic, which predominates at night [14,15,16].

Circadian alterations in peripheral temperature have been associated with dysautonomia in ME/CFS, reflecting alterations in the vasoconstriction/vasodilation process [13]. The peripheral temperature increases at night due to vessel dilation, in order to favor sleep, while alterations in this process have been related to difficulty in sleep onset [17]. However, vasoconstriction is due not only to the sympathetic effect on the vessels, but also to the regulation of local vessels by endothelial function, which has been reported to be altered in ME/CFS [18,19,20].

Endothelin-1 (ET-1), a strong endothelial vasoconstrictor, has been related to the regulation of the circadian rhythms [21], and high circulating ET-1 levels have recently been reported in ME/CFS [19]; thus we hypothesized that there would be an alteration of the vasoconstrictor/vasodilation process, which might be reflected in temperature variations, and would indicate dysautonomia in ME/CFS.

In this study, we aimed to explore the autonomic responses of ME/CFS patients, through the study of hemodynamic variables and thermoregulation, and their association with the endothelium state. Thus, we tested the utility of a passive standing test (10-min NASA lean test, NLT) [22] to assess orthostatic intolerance (OI) and also we investigated peripheral temperature changes in the study participants by (a) measuring the wrist temperature rhythms through actigraphy (i.e., the temperature variation between day and night) and (b) recording the skin temperature changes when participants changed from a supine to standing position during the NLT. Moreover, we also explored the functioning endothelial status through the measurement of circulating endothelial biomarkers in the study participants.

## 2. Results

### 2.1. Demographic and Clinical Characteristics of Study Population

Table 1 summarizes the demographic and clinical characteristics and the results of routine blood testing of the participants. The groups differed in terms of age (*p* = 0.016) and BMI (*p* < 0.001). In addition, ME/CFS patients showed higher levels of cholesterol, triglycerides (TG), and low-density lipoproteins (LDL) (all *p* < 0.01), and also of the hormone 17β-estradiol (*p* = 0.044), compared to healthy controls, although the values of this last variable were inside the normal range. We did not control for the menstrual cycle stage among participants; however, healthy volunteers were younger than ME/CFS patients (37.5% vs. 50.5% were aged over 50). No statistically significant differences were reported for menopause between ME/CFS and healthy controls (34% vs. 31%). Thus, since there is a strong association of LDL, cholesterol, and TG with age (*p* < 0.05 for all), and also between TG and BMI (*p* < 0.005), the comparison of the other variables displayed in Table 2, and the further statistical tests, were always carried out adjusting for age and BMI.

Additionally, the groups differed in the self-reported outcome measures, with ME/CFS patients recording higher scores for fatigue severity, anxiety/depression symptoms, sleep quality, autonomic symptoms, and lower health-related quality of life (*p* < 0.001 for all).

### 2.2. Passive Standing Test

All participants finished the orthostatic test (10-min NLT). Blood pressure and HR values were significantly higher in ME/CFS than in healthy controls; however, no differences according to time of the day were observed. As for changes in hemodynamic variables, there were no differences according to groups (Table 2).

Seven ME/CFS patients (four in the morning and three in the afternoon) and eight healthy controls (six in the morning and two in the afternoon) had POTS. Moreover, four ME/CFS patients and one healthy control had OH; thus, 16% of ME/CFS patients (11/67) and 13% of healthy controls (6/48) had abnormal cardiovascular responses to position changes after NLT, with no differences between the two groups in terms of distribution. However, although no differences were found between groups in the postural autonomic responses, ME/CFS patients had higher values of BP and HR in both the supine and standing position, and in individuals with and without POTS (*p* < 0.05 in all ANOVA comparisons) (Figure 1a–c).

### 2.3. Postural Wrist Temperature Changes

Postural wrist temperature-related changes (WT) could only be measured in 42 ME/CFS patients (15 in the morning and 27 in the afternoon) and in 33 healthy controls (17 in the morning and 16 in the afternoon). Mean WT varied according to the time of the day, and was always higher in the afternoon (WT morning: 28.5 ± 0.37 °C and WT afternoon: 31.5 ± 0.27 °C), but no differences were found between the groups (Table 2). Wrist temperature rose significantly during the NLT (*p* < 0.01 in both ME/CFS and healthy controls), the increase being higher in the morning (mean value 2.35 ± 0.23 °C vs. 0.76 ± 0.22 °C in the afternoon), with no differences between ME/CFS patients and healthy controls.

Changes in WT during the NLT (ΔWT_NLT) were positively associated with nocturnal temperature (T_M5, r = 0.430; *p* < 0.007), and with variables related to the stability of the WT circadian rhythm, described in the Section 4 (T_R, r = 0.417; *p* = 0.012, and T_PV, r = 0.409; *p* = 0.014) in ME/CFS, but these associations were not present in controls. In addition, in patients, ΔWT_NLT was associated with hemodynamic variables such as SBP (r = 0.350; *p* = 0.031) and DBP (r = 0.371; *p* = 0.028) and negatively with the decrease in DBP during the NLT. In healthy controls, these associations were not observed. ΔWT_NLT was negatively associated with ET-1 levels both in controls (r = −0.676; *p* = 0.006) and in ME/CFS (r = −0.635; *p* = 0.034). However, it should be remembered that these two variables could only be measured together in a few individuals (15 ME/CFS and 17 healthy controls).

### 2.4. Wrist Temperature Rhythms and Motor Activity Measured by Actigraphy

Circadian rhythm shows similar profiles in the two groups (Figure 2a,b). However, once corrected for age and BMI, an ANOVA indicated that variables obtained from activity data differed between groups, but not those obtained from temperature data (Table 2). Specifically, maximum activity, mean daily activity, and the amplitude of the rhythm were lower in ME/CFS. Body mass index emerged as the most important factor for WT rhythm variables.

We also tested the association between the PSQI questionnaire (sleep alterations) and T_M5, since the nocturnal temperature increases with sleep. As expected, in controls, the correlation was negative, but in ME/CFS, it did not reach significance. Moreover, since both activity and WT circadian rhythms tend to be associated, we tested whether this association occurred in a similar way in both groups, finding that, in controls, the nocturnal temperature (T_M5) was associated with high daily motor activity (r = 0.433; *p* = 0.003), high rhythm amplitude (r = 0.452; *p* = 0.002), and stability of the activity rhythm (r = 0.318; *p* = 0.033); these associations were not found for ME/CFS.

### 2.5. Endothelial Function Biomarkers

As shown in Table 2, ME/CFS patients showed significantly higher levels of plasma ET-1 and VCAM-1 proteins than healthy controls (*p* < 0.05 adjusted for both age and BMI). However, no differences were found for the ICAM-1 protein. To relate endothelial dysfunction to the rest of the variables, regression models with ET-1, VCAM-1, or ICAM-1 as dependent variables were carried out considering the following variables as predictors: (a) those obtained from temperature rhythms, (b) those obtained from motor activity rhythms, and (c) those obtained from the NLT.

Table 3 shows the model of the selection process with the final coefficients and their statistical significance. Specifically, our results showed the following.

Circadian variables of WT were significant predictors of ET-1 in ME/CFS and healthy controls. As such, the final model was obtained with the stability of the circadian rhythm, amplitude, and nocturnal value (T_M5). Meanwhile, in controls, it was the amplitude of the WT circadian pattern that was the significant predictor of ET-1.None of the circadian variables of motor activity were predictors of ET-1 in ME/CFS, nor in healthy controls. As such, no variables remained in the final model.NLT variables were significant predictors of ET-1 in healthy controls. In the final model, ET-1 was associated with SBP/DBP, ΔSBP, and ΔDBP in healthy controls. This association was not found in ME/CFS patients.

Interestingly, VCAM-1 and ICAM-1 were not associated with the temperature variables. Also of interest, ET-1 levels were positively associated with the self-reported outcome measures in ME/CFS, but not in healthy controls; the higher the ET-1 levels, the worse the symptomatology (Table 4).

### 2.6. Discriminant Analysis

Finally, we tested the association of the subjective perception of the autonomic symptoms with the objective variables. To do so, we carried out a stepwise regression analysis with the global COMPASS-31 score as a dependent variable, and the representative and non-redundant variables for each group as predictors: namely, the hemodynamic variables (DBP, HR, ΔSBP_3L, ΔDBP_3L, and ΔHR_3L), temperature circadian rhythm variables (amplitude, T_PV, and T_M5), and endothelial biomarkers (ET-1 and VCAM-1). Again, a stepwise linear regression model was applied and collinearity was analyzed (Table 5). 

The results indicate that the predictors for the best model with COMPASS-31 as the dependent variable were DBP, ET-1, VCAM-1, T_PV, and T_M5. A MANOVA was followed up with discriminant analysis of these variables, which revealed significant differences between the two groups (Wilks’ lambda = 0.372; Chi-square = 53.95; *p* < 0.0001). In all, 91.5% of participants of the original grouped cases (96% of ME/CFS patients and 85% of healthy controls) were correctly classified.

## 3. Discussion

This work is the first to study indicators of autonomic dysfunction through variables related to the vasoconstriction/vasodilation process, also considering the subjective symptomatology of ME/CFS patients. The main outcome of this study is the association of circulating ET-1 levels with the self-reported frequency/severity of autonomic symptoms and WT-related variables, which suggests a potential role of endothelial dysfunction in ME/CFS [19]. Moreover, a discriminant analysis using blood pressure, endothelium biomarkers, and circadian variables allows a good classification of the participants into patient and control groups. This indicates that only a multi-functional approach could shed light on the complexity of autonomic symptoms in ME/CFS. 

This study also addresses secondary questions such as the usefulness of NLT, the measurement of peripheral WT, and the analysis of alterations of circadian rhythms in ME/CFS. The inclusion of NLT in the diagnostic evaluation is to add objective values to the autonomic symptoms in ME/CFS patients, in addition to completing the COMPASS-31 and OGS measures. However, in agreement with Roerink et al. [12], we did not find differences in OI responses between groups; nor could we correlate the hemodynamic response with the symptom-related questionnaires. In any case, one should bear in mind that the diagnosis of POTS is not based on increases in HR alone, since other OI criteria should also be included [11]. Our results confirmed that ME/CFS patients had higher values of BP and HR than matched healthy controls, independently of the orthostatic symptoms, corroborating the idea that the ME/CFS condition may induce a circulatory decompensation [22]. Increased HR in the supine position has already been reported in ME/CFS and may reflect increased sympatho-adrenomedullary activity [22,23]. Although most ME/CFS patients were not hypertensive, the increase in BP could be related to arterial stiffness, which has also been described in ME/CFS and in turn has been associated with more sympathetic dominance [24,25].

To our knowledge, this is the first study to include the measurement of WT during the NLT. Postural skin temperature changes may be due to blood redistribution and may reflect the individual variability of vasomotor activity [26]. Essentially, the skin (primarily distal site) may serve as an ideal reservoir for blood redistribution due to changes in skin vascular tone [27]. With these considerations in mind, we assumed that the skin temperature would rise in the upright position, partially due to gravity, which favors the presence of blood in the lower part of the arm, but also due to the vasodilator response.

Although we did not find differences in skin temperature between groups during NLT, this variable was of interest since it may be associated with the amplitude of the WT circadian rhythm and with ET-1 concentrations and may suggest a different thermoregulatory response in ME/CFS. However, since postural changes in WT and ET-1 levels could only be measured together in a relatively low number of individuals, further studies are needed to analyze this issue in greater depth.

A previous study by our group found significant differences in the activity rhythm between ME/CFS and healthy controls, with ME/CFS showing less activity during the day, a lower amplitude, and more fragmentation [3]. In agreement with this earlier study, on this occasion, we did not find differences in the WT rhythm. Nevertheless, only WT circadian rhythm variables were predictors of soluble ET-1 levels, a finding that corroborates the association between these variables. Thus, the fact that the soluble ET-1 protein might be correlated with the variables of the WT circadian rhythm but not with variables of the motor activity circadian rhythm in ME/CFS indicates that ET-1 and skin temperature are both markers of vasoconstriction/vasodilation processes, which may be impaired in ME/CFS and may be related to the intrinsic endothelial structure of the blood vessel [28]. Thus, our study points to endothelial dysfunction as a prominent feature that may help to explain the autonomic symptoms in ME/CFS; in our view, measuring the skin temperature might provide useful information for the clinical evaluation in ME/CFS.

On the other hand, endothelial damage alters the balance between vasoconstrictor/vasodilation events and also expresses higher levels of adhesion molecules. In our cohort, VCAM-1 levels were also high in ME/CFS, suggesting endothelial inflammation; however, the levels did not correlate with the severity of the symptomatology. Interestingly, levels of the vasoconstrictor ET-1 marker were elevated in ME/CFS, as other authors have recently reported [19]; since this is the variable that best correlated with the results of the self-reported questionnaires, it could be an indicator of illness severity. However, one must take into account that ME/CFS patients also had high levels of cholesterol, triglycerides, and LDL which may have caused endothelial dysfunction and cardiovascular disease, and may play a crucial role in the pathogenesis of ME/CFS [29]. In addition, none of the ME/CFS patients were taking contraceptive pills; however, mean estrogen levels were higher in ME/CFS patients (menopause: 34%), although the mean age was higher in this group. Further clinical and experimental studies are required to assess the sex hormone imbalance and autonomic dysfunction in people with ME/CFS.

Interestingly, sleep perception in healthy controls (as reflected by PSQI scores) was associated with nocturnal WT, an objective variable that increased during sleep due to vasodilation. However, this association was not found in ME/CFS patients, who often report unrefreshing sleep. This may be due to the autonomous nervous system dysfunction in our ME/CFS cohort, or to a decrease in circulating levels of melatonin, a key regulatory hormone of body temperature rhythm and sleep, which also affects autonomic vagal activity. In fact, sleep disturbances have been associated with elevated BP and HR and with lower HR variability, indicating reduced parasympathetic activity at night [30]. Since sleep and the cardiovascular function are regulated by the networks of central nervous system nuclei, their simultaneous study is also important in determining further relationships between sleep–wake control circuitry and the pathways regulating the autonomic function [31].

### Limitations of the Study

The present study has some limitations that should be mentioned. First, the single-center, cross-sectional nature of the design prevents us from identifying causation. In addition, the relatively small sample size, the use of self-reported data, and the inclusion only of women with mild/moderate disease severity may mean that the results are not representative of the entire population. Further limitations include the lack of data on comorbid health conditions, menstrual cycle stage, metabolic syndrome factors, physical activity, lack of endothelial function assessment in the clinical setting, and other lifestyles that may independently explain the endothelial dysfunction in the participants. Finally, the analyses derived from ET-1 associations are exploratory and may not have been appropriately powered.

## 4. Materials and Methods

### 4.1. Study Participants

A single-center, prospective, cross-sectional case–control study of 67 consecutive females with ME/CFS and 48 non-fatigued healthy controls recruited at a single outpatient tertiary-referral center (ME/CFS clinical unit, Vall d’Hebron University Hospital, Barcelona, Spain) from October 2019 to March 2022 was conducted. Following the suggestion made in our previous study, data were not recorded during the summer [3]. After receiving verbal and written information on the study protocol, each study participant gave signed informed consent to participate prior to enrollment, which was approved by the Vall d’Hebron Hospital Institutional Review Board (reference number PR/AG 201/2016). The study was carried out in accordance with the principles set in the declaration of Helsinki and with all the international literature on harmonization and good clinical practice guidelines.

Patients with ME/CFS were potentially eligible if they were female, aged ≥ 18 years, with a confirmed diagnosis by a specialist of ME/CFS according to the 2011 international consensus criteria [7]. Healthy control subjects were eligible if they were adult females and neither met the case criteria for ME/CFS nor reported orthostatic intolerance symptoms at the time of study inclusion. Healthy controls were recruited through word-of-mouth from the local community. None of the participants were taking contraceptive pills.

Participants were subjected to stringent exclusion criteria, as previously described by our group [3]. The major exclusion criteria were a previous or current diagnosis of an autoimmune disorder, multiple sclerosis, psychosis, major depression disorder, heart disease, hematological disorders, infectious diseases, sleep apnea or thyroid-related disorders; pregnancy or breast-feeding; smoking habit; strong hormone-related drugs; and fatigue-associated symptoms that did not conform to the ME/CFS case criteria used for this study. Demographic and clinical characteristics of the study population are displayed in Table 1.

### 4.2. Experimental Procedures

Thirty-two ME/CFS patients and 29 healthy controls came to the local hospital on Wednesday morning between 8 a.m. and 11 a.m. and another cohort (35 ME/CFS patients and 19 healthy controls) on Tuesday afternoon between 3 p.m. and 6 p.m. for a clinical assessment. Demographic and self-reported outcome measures were recorded, as briefly described below and detailed in our previous study [3]. In participants who attended in the morning, a blood sample was taken for routine biochemical analysis. All participants underwent the same orthostatic test protocol (10-min NLT) to evaluate orthostatic intolerance (see details below), and wrist temperature (WT) was also recorded using a temperature sensor (Thermochron iButton^®^ DS1921H, San Jose, CA, USA) placed on the right wrist of each participant during the NLT procedures. 

Participants were asked to wear an ambulatory actigraphy device (ActTrust^®^, Condor instruments, Sao Paulo, Brazil) on the wrist of the non-dominant arm over the radial artery continuously for seven days, except when showering or at the swimming pool. The same actigraphy was programmed to collect data on activity (arbitrary units), skin temperature (°C), and light intensity (lux) at one-minute intervals. The data were recorded and stored in the device’s memory for further analysis. Subjects returned to the hospital after one week to hand in the actigraphy device, and completed health-related questionnaires.

### 4.3. Measures

Participants were also asked to provide complete validated self-report questionnaires on their current health status one week after the first clinical assessment. Changes in fatigue perception (FIS-40) [32], sleep quality (PSQI) [33], anxiety/depression (HADS) [34], autonomic symptoms (COMPASS-31) [35], and health-related quality of life (SF-36) [36] were assessed as described in our previous study [3], except for the assessment of the frequency and severity and interference of orthostatic symptoms, which was performed using the orthostatic grading scale (OGS).

### 4.4. Orthostatic Grading Scale

The orthostatic grading scale (OGS) is a 5-item validated self-reported questionnaire designed to assess symptoms of orthostatic intolerance due to orthostatic hypotension. The five questions address the frequency/severity and interference of orthostatic symptoms in daily life activities. Respondents rate each item on a scale of 0 to 4. Adding the scores for the individual items produces an overall OGS score ranging from 0 (never or rarely orthostatic symptoms) to 20 (maximum orthostatic symptoms). Higher scores indicate greater severity of autonomic dysfunction [37].

### 4.5. Assessment of Cardiovascular Autonomic Function

Autonomic response was evaluated using a passive standing test (10-min NASA lean test, NLT), a simple and well-established non-invasive procedure used to assess impaired cardiovascular compensatory responses to standing in ME/CFS. The NLT classifies OI phenotypes as orthostatic hypotension (OH) and postural orthostatic tachycardia syndrome (POTS), by measuring hemodynamic parameters, blood pressure (BP), and heart rate (HR) for both clinical and research purposes. The test was conducted in a consistent manner by the same examiner in the morning or in the afternoon, in a quiet room with an average relative temperature of 22.1 ± 1.2 °C and humidity of 55 ± 5.8%.

The participants were first asked to lie down for five minutes and then to stand and lean against a wall, with heels 6–8 inches away from the wall. An automated BP cuff with a monitor (Beurer BM-26, Beurer GmbH & Co., Ulm, Germany) was placed on the left arm, recording the systolic blood pressure (SBP) and diastolic blood pressure (DBP) and HR at 1-min intervals, and, simultaneously, a temperature sensor (Thermochron iButton^®^ DS1921H, San Jose, CA, USA) was placed on the right wrist to record the peripheral temperature changes throughout the orthostatic test. SBP, DBP, and HR were recorded every minute for the two last minutes in the supine position and during the full 10-min after attaining the upright position. Throughout the recording, participants were asked to remain still, and any talking or movement was discouraged, except for reporting any symptoms of concern.

The NLT was stopped early at the request of the subject, or in the event of severe pre-syncope [22]. After 10 min upright, each participant was asked about the frequency/severity and impact of orthostatic symptoms on a 5-item OGS score [37].

### 4.6. Concomitant Medication

In order to maintain patients’ functional status, usual medication was not withdrawn during the study. However, information on current medication use was collected from the medical records. Thirty-nine patients (58%) were taking non-steroidal anti-inflammatory drugs (ibuprofen, celecoxib for generalized pain), twenty-six (39%) serotonin-norepinephrine reuptake inhibitors (duloxetine for neuropathic pain), twelve (18%) anticonvulsants (gabapentin, pregabalin for chronic neuropathic pain), thirteen (19%) analgesics (paracetamol), and thirty-four (51%) anxiolytics (alprazolam, quetiapine for generalized anxiety disorder). Only seven patients (10%) were taking antihypertensive drugs, and none on the day of the NLT. None of the healthy controls were taking any medication.

### 4.7. Hemodynamic Definitions Recorded during the Orthostatic Test

Criteria for OI were based on the 2021 expert consensus statement and guidelines on the definition of OH and POTS as follows: (a) orthostatic hypotension was defined as a decrease in SBP of ≥20 mmHg, or a decrease in DBP of ≥10 mmHg in the first three minutes standing compared with resting supine values; and (b) POTS was defined as either an increase in HR ≥ 30 bpm and/or a current HR ≥ 120 bpm based on the average of the last three minutes standing [38,39]. Classification of OI (OH and POTS) during the 10-min NLT was quantified as the difference between supine and standing hemodynamic changes (ΔBP and ΔHR), as previously described [22]. For the purposes of this study, SBP/DBP and HR were used as raw values recorded during the NLT. For correlation analysis and grouping comparison, we calculated the mean values during the last two minutes in the supine position (sp), and the mean values during first three minutes standing (3F), and mean values of minutes 8–10 standing (last three minutes, 3L). The changes in these variables compared with the values in the supine position were also calculated (Δ3L or Δ3L).

### 4.8. Actigraphy Analysis

Skin temperature and activity data obtained with the actigraphy device were analyzed with “El-temps, version 314”, an integrated package for chronobiological analysis (Prof. A. Díez-Noguera, University of Barcelona, Spain; https://www.el-temps.com) (accessed on 15 September 2022). The rhythmic variables were determined by adjusting the data to a 24-h co-sinusoidal curve. Thus, data of each variable (skin temperature and activity) were calculated: the mean 24-h value (MESOR), the acrophase (time of the day when the maximum value of the variable occurs), and the amplitude of an adjusted 24-h sinusoidal rhythm. Non-parametric circadian analysis was also performed as previously described [40]: M10 (or M5 in the case of WT) and L5 (or L10 in the case of WT) denoted the mean temperature in the 10 (or 5) consecutive hours with highest values (night temperature or activity during the day) and the 10 (or 5) hours with lowest values (day temperature or night activity values), respectively. Note that activity increases during the day and the skin temperature during the night. In addition, the intra-daily variability (IV), the stability of the rhythm measured by the grouping of the acrophases (Rayleigh’s test, R), and the percentage of variance explained by the 24-h rhythm (PV) were also calculated. To facilitate comprehension, when these variables were obtained from temperature data, the name of the variables starts with T, and when obtained from activity, they start with A.

### 4.9. Blood Sampling and Processing

A total of twenty milliliters of fasting blood samples from each participant were collected into K_2_EDTA anticoagulant tubes (BD vacutainer, Becton, Dickinson and Company, ON, Canada) from an antecubital vein with a 19-gauge needle without venous stasis. One tube was transported to the local core laboratory for the assessment of routine blood tests, including a comprehensive metabolic panel. All other blood samples were immediately centrifuged at 2500 rpm for 15 min at 4 °C (Thermo Scientific, Waltham, MA, USA) and then supernatants were collected and stored in aliquots at −80 °C until assayed. No sample was thawed more than twice. Repeated samples from each participant were measured in the same analytical batch.

### 4.10. Measurement of Endothelial Biomarkers

Circulating levels of soluble ET-1, vascular cell adhesion molecule-1 (VCAM-1), and intracellular adhesion molecule-1 (ICAM-1) were measured as indicators of endothelial functioning status. Plasma concentrations of ET-1 (cat n° DET-100), VCAM-1 (cat n° DVC00), and ICAM-1 (cat n° DCIM00) proteins were assayed in each participant using commercially available ELISA kits according to the manufacturer’s instruction manual (Quantikine R&D Systems, Minneapolis, MN, USA), using a Synergy™ H1M, hybrid multi-mode microplate reader (BioTek Instruments, Inc., Winooski, VT, USA) at O.D. 450 nm. Results were analyzed by comparison with standard calibration curves in each well, and are presented as averages of two duplicated samples.

### 4.11. Statistical Analysis

Data were tested for normality and homogeneity of variance with the Shapiro–Wilk and Levene tests, respectively. Data are presented as mean ± standard error of means (SEM) for continuous variables. Statistically significant differences for parametric variables were tested by means of one-way ANOVA. Since the NLT was conducted during the morning in some participants and during the afternoon in others, in the analysis of NLT variables, the time of the day was also included as an independent factor. In cases of non-parametric variables such as those obtained in the questionnaires, the Mann–Whitney U test was assayed to test differences between groups. Meanwhile, differences in categorical variables were analyzed using Fisher’s exact test. Pearson’s correlation analyses were used in paired data to evaluate the associations between the different variables. For each variable studied, paired data with missing values were excluded from the analysis. In addition, stepwise linear regression models with backwards elimination were performed to select significant predictors of endothelial biomarkers (ET-1, VCAM-1, or ICAM-1). Subsequently, we tested the interaction between the autonomic symptoms assessed by COMPASS-31 and the significant predictors of endothelial biomarkers using stepwise linear regression models. Finally, we conducted a discriminant analysis using the variables obtained in the last analysis to evaluate which were the variables that could reliably classify the subjects into the two groups. Univariate F-tests were then calculated to determine the importance of each independent variable in forming the discriminant functions. Examining the Wilk’s lambda values for each of the predictors revealed how important the independent variable was to the discriminant function, with smaller values representing greater importance. Data were analyzed using IBM SPSS Statistics for Windows, version 27.0 (IBM Corp., Armonk, NY, USA). All analyses were adjusted for age and BMI. *p*-values ≤ 0.05 was considered statistically significant.

## 5. Conclusions

This is the first study reporting indicators of the vasoconstriction and vasodilation process, such as peripheral skin temperature variations and vascular endothelium function biomarkers, considering all participants’ self-reported symptomatology of ME/CFS. Taken together, future studies in this area should aim to provide a better characterization of the underlying mechanisms of dysautonomia and endothelial dysfunction, which may contribute to the development of new biological targets for potential therapeutic interventions in ME/CFS.

## Figures and Tables

**Figure 1 ijms-24-04835-f001:**
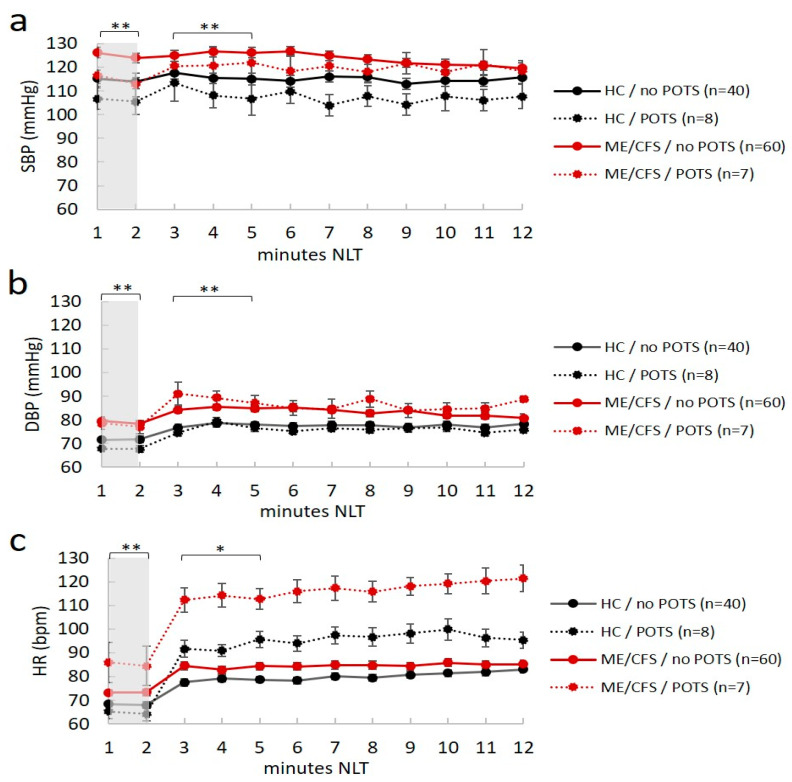
Mean values of hemodynamic responses (blood pressure and heart rate) plotted during the passive orthostatic test (10 min NASA lean test) in the first two minutes (resting supine position highlighted in gray) and the following 10 min (standing up position from three to 12 min) in healthy controls (HC) and ME/CFS with and without POTS (panel (**a**–**c**)). Data are shown as mean ± SEM. Data were analyzed by ANOVA with raw data (see text). Significance was set at * *p* < 0.05 and ** *p* < 0.01, indicating differences between ME/CFS and HC in the no POTS group. No difference was reported for ME/CFS and healthy controls in the POTS group. SBP, systolic blood pressure (mmHg); DBP, diastolic blood pressure (mmHg); HR, heart rate (bpm). ME/CFS patients always had statistically higher hemodynamic values (BP and HR) than healthy controls at rest (*p* < 0.012).

**Figure 2 ijms-24-04835-f002:**
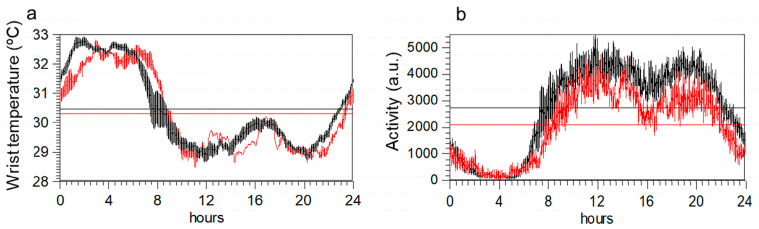
Mean 24-h circadian waveforms for wrist temperature (panel (**a**)) and motor activity (panel (**b**)) in ME/CFS patients (red) and healthy controls (black). Data represent the mean daily profile on 7-day actigraphy records. Horizontal lines in each graph indicate the maximum and minimum values of each profile.

**Table 1 ijms-24-04835-t001:** Baseline demographic and clinical characteristics of the study population.

Variables ^a^	ME/CFS	HCs	Adjusted *p*-Value ^b^
	(n = 67)	(n = 48)	
Age, years	49.0 ± 1.04	44.3 ± 1.90	**0.016**
BMI, kg/m^2^	27.6 ± 5.50	24.1 ± 0.60	**<0.001**
Supine SBP, mmHg	124.18 ± 1.9	112.95 ± 1.6	**<0.05**
Supine DBP, mmHg	78.97 ± 1.10	70.97 ± 1.06	**<0.01**
Supine HR, bpm	74.47 ± 1.44	67.5 ± 1.18	**<0.01**
Duration of illness, years	6.21 ± 0.50	n/a	n/a
Routine blood tests ^§^			
Hematocrit, %	42.8 ± 0.82	42.4 ± 0.63	0.667
Glucose, mg/dL	89.1 ± 1.86	85.30 ± 1.58	0.122
Urea, mg/dL	31.2 ± 1.48	29.5 ± 1.47	0.406
Creatinine, mg/dL	0.69 ± 0.03	0.73 ± 0.02	0.310
Urate, mg/dL	4.69 ± 0.24	4.48 ± 0.19	0.493
Cholesterol, mg/dL	227.07 ± 7.87	200.33 ± 7.30	**0.016**
Triglycerides, mg/dL	113.90 ± 10.67	71.78 ± 4.86	**0.001**
HDL, mg/dL	66.86 ± 2.87	69.24 ± 3.19	0.582
LDL, mg/dL	141.61 ± 6.42	118.76 ± 6.05	**0.012**
eGFR, mL/min/1.73 m^2^	72.6 ± 7.22	76.0 ± 2.00	0.790
Sodium, mmol/L	139.75 ± 0.31	138.95 ± 0.29	0.060
Potassium, mmol/L	4.17 ± 0.06	4.22 ± 0.05	0.503
Phosphate, mg/dL	3.59 ± 0.12	3.49 ± 0.08	0.478
Calcium, mg/dL	9.66 ± 0.07	9.65 ± 0.05	0.973
Albumin, g/dL	4.35 ± 0.05	4.46 ± 0.05	0.137
Total protein, g/dL	7.14 ± 0.07	7.26 ± 0.08	0.284
TSH, mUI/L	2.12 ± 0.20	2.01 ± 0.23	0.715
Free T4, ng/dL	1.10 ± 0.20	1.18 ± 0.03	0.071
25-hydroxy-vitamin D, ng/mL	20.74 ± 2.08	23.50 ± 1.59	0.293
Cortisol, μg/dL	14.96 ± 1.18	12.61 ± 1.28	0.183
17β-estradiol, pg/mL	118.23 ± 27.65	52.07 ± 16.42	**0.044**
Progesterone, ng/mL	0.54 ± 0.25	0.72 ± 0.51	0.847
Aldosterone, ng/dL	10.35 ± 0.98	10.03 ± 0.51	0.772
Prolactin, ng/mL	8.53 ± 0.62	8.38 ± 0.58	0.864
Measures ^a^			
FIS-40			
Global score (0–160)	130.3 ± 2.7	18.9 ± 3.0	**<0.001**
Physical	33.9 ± 0.6	82.6 ± 0.8	**<0.001**
Cognitive	61.9 ± 1.5	8.6 ± 1.5	**<0.001**
Psychosocial	34 ± 0.8	4.5 ± 0.8	**<0.001**
PSQI			
Global score (0–21)	15.5 ± 0.5	5.9 ± 0.5	**<0.001**
Subjective sleep quality	2.2 ± 0.11	0.7 ± 0.11	**<0.001**
Sleep latency	2.1 ± 0.31	0.6 ± 0.16	**<0.001**
Sleep duration	1.9 ± 0.11	1.1 ± 0.09	**<0.001**
Habitual sleep efficiency	2.1 ± 0.14	0.6 ± 0.14	**<0.001**
Sleep disturbances	2.3 ± 0.18	1.1 ± 0.05	**<0.001**
Use of sleeping medication	2.1 ± 0.15	0.3 ± 0.11	**<0.001**
Daytime dysfunction	2.4 ± 0.08	0.6 ± 0.97	**<0.001**
HADS			
Global score (0–42)	24.5 ± 0.9	6.9 ± 0.7	**<0.001**
Anxiety	12.8 ± 0.5	5.3 ± 0.5	**<0.001**
Depression	11.7 ± 0.5	1.7 ± 0.32	**<0.001**
COMPASS-31			
Global score (0–100)	59.5 ± 1.68	13.9 ± 1.53	**<0.001**
Orthostatic intolerance	25.4 ± 0.9	5.9 ± 0.9	**<0.001**
Vasomotor	1.9 ± 0.18	0.1 ± 0.07	**<0.001**
Secretomotor	10.3 ± 0.32	1.4 ± 0.37	**<0.001**
Gastrointestinal	13.5 ± 0.50	4.8 ± 0.57	**<0.001**
Bladder	4.4 ± 0.35	0.6 ± 0.17	**<0.001**
Pupillomotor	3.8 ± 0.12	1.06 ± 0.13	**<0.001**
SF-36			
Global score (0–100)	26.0 ± 3.1	85.5 ± 2.5	**<0.001**
Physical functioning	30.3 ± 2.5	96.9 ± 0.8	**<0.001**
Physical role functioning	1.8 ± 0.9	90.5 ± 3.3	**<0.001**
Bodily pain	20.9 ± 2.9	85.4 ± 2.9	**<0.001**
General health perception	23.1 ± 1.9	85.6 ± 1.7	**<0.001**
Vitality	13.5 ± 1.9	70.6 ± 2.7	**<0.001**
Social role functioning	34.4 ± 3.1	92.7 ± 2.2	**<0.001**
Emotional role functioning	39.4 ± 5.6	87.4 ± 4.1	**<0.001**
Mental health	43.9 ± 2.8	75.2 ± 2.5	**<0.001**
OGS ^†^			
Global score (0–20)	12.82 ± 4.39	0.87 ± 1.81	**<0.001**
Frequency of orthostatic symptoms	2.68 ± 1.04	0.26 ± 0.59	**<0.001**
Severity of orthostatic symptoms	2.59 ± 1.01	0.22 ± 0.47	**<0.001**
Conditions under which orthostatic symptoms occur	2.73 ± 1.12	0.24 ± 0.48	**<0.001**
Interference with activities of daily living	2.50 ± 1.06	0.02 ± 0.15	**<0.001**
Standing time	2.32 ± 1.21	0.12 ± 0.55	**<0.001**

Data are shown as mean ± SEM for each item. ^a^ Baseline self-reported outcome measures as explained in Methods section. ^b^ *p*-values from Student’s *t*-test for continuous variables and from Mann–Whitney U-test for categorical variables. Bold values denote statistical significance at *p* < 0.05 between groups. ^§^ Thirty ME/CFS cases and 29 healthy controls underwent fasting laboratory blood tests. ^†^ Twenty-nine ME/CFS cases and 16 healthy controls completed the OGS questionnaire. ME/CFS, myalgic encephalomyelitis/chronic fatigue syndrome; HCs, healthy controls; BMI, body mass index; SBP, systolic blood pressure; DBP, diastolic blood pressure; n/a, not applicable; HDL, high-density lipoproteins; LDL, low-density lipoproteins; eGFR, estimated glomerular filtration rate; TSH, thyroid-stimulating hormone; FIS-40, 40-item fatigue index scale; HADS, hospital anxiety and depression scale; PSQI, Pittsburgh sleep quality index; COMPASS-31, 31-item abbreviated composite autonomic symptom score; SF-36, medical outcome study 36-item short form health survey; OGS, orthostatic grading scale.

**Table 2 ijms-24-04835-t002:** Differences between cardiovascular variables, wrist temperature, circadian-related variables, and endothelial biomarkers in ME/CFS patients and healthy controls.

Variables	ME/CFS (n = 67)	HCs (n = 48)	Adjusted *p*-Value ^a^
Cardiovascular variables			
Supine SBP, mmHg	124.18 ± 1.99	112.95 ± 1.6	**<0.012**
ΔSBP_3F, mmHg	1.06 ± 1.14	1.79 ± 1.01	0.979
Supine DBP, mmHg	78.97 ± 1.13	70.97 ± 1.06	**<0.001**
ΔDBP_3F, mmHg	6.69 ± 0.84	6.66 ± 0.87	0.959
Supine HR, bpm	74.47 ± 1.44	67.5 ± 1.18	**<0.001**
ΔHR_3L, bpm	14.65 ± 1.23	17.28 ± 1.64	0.961
Wrist temperature (WT) during the NLT ^†^			
Supine WT, °C	30.72 ± 0.32	29.67 ± 0.39	0.474
ΔWT_3L, °C	1.08 ± 0.25	1.97 ± 0.28	0.173
Temperature rhythm variables			
T_Mesor, °C	30.35 ± 0.10	30.54 ± 0.12	0.745
T_Acrophase, min	318.77 ± 33.19	190.55 ± 12.91	0.391
T_Amplitude, °C	1.79 ± 0.10	1.73 ± 0.11	0.381
T_IV, a.u.	0.01 ± 0	0.01 ± 0	0.116
T_R, a.u.	0.83 ± 0.03	0.88 ± 0.02	0.423
T_ PV, %	41.15 ± 2.03	42.07 ± 1.56	0.887
T_M5, °C	32.52 ± 0.15	32.82 ± 0.12	0.621
T_L10, °C	28.96 ± 0.11	29.28 ± 0.17	0.259
Activity rhythm variables			
A_ Mesor, a.u.	2058 ± 95	2746 ± 82	**<0.001**
A_Acrophase, min	858 ± 11.4	856 ± 11	0.986
A_ Amplitude, °C	1744 ± 80	2116 ± 72	**<0.009**
A_ IV, a.u.	0.42 ± 0.008	0.37 ± 0.01	0.102
A_ R, a.u.	0.93 ± 0.009	0.92 ± 0.009	0.745
A_ PV, %	32.73 ± 0.97	31.02 ± 0.88	0.107
A_ M10, a.u.	3390 ± 149	4308 ± 127	**<0.001**
A_L5, a.u.	199 ± 26	175 ± 20	0.911
Endothelial biomarkers ^§^			
ET-1, pg/mL	1.84 ± 0.14	1.12 ± 0.63	**0.004**
VCAM-1, ng/mL	739 ± 33	494 ± 17	**< 0.001**
ICAM-1, ng/mL	284 ± 10	285 ± 19	0.611

Data are expressed as mean ± SEM. ^a^ *p*-values were attained from ANOVA, which was adjusted for age and BMI. Significant comparisons are highlighted in bold. ^†^ Wrist temperature was recorded during the NLT in 42 ME/CFS cases and 33 healthy controls. ^§^ Circulating endothelial biomarkers were measured in 32 ME/CFS cases and 29 healthy controls. ME/CFS, myalgic encephalomyelitis/chronic fatigue syndrome; HCs, healthy controls; SBP, systolic blood pressure; ΔSBP_3F, variation in SBP in the first three minutes of standing; DBP, diastolic blood pressure; ΔDBP_3F, variation in SBP in the first three minutes of standing; HR, heart rate; ΔHR_3, variation in HR in the last three minutes of standing; IV, intraday variability; R, Rayleigh test; PV, variance (%) explained by the 24-h rhythm; M10 (or M5), maximum value of the variable; L5 (or L10), minimum value of the variable (see text for further explanation); ET-1, endothelin-1; VCAM-1, vascular cell adhesion molecule-1; ICAM-1, intercellular adhesion molecule-1. Variables starting with T_ or A_ were obtained, respectively, from the temperature or activity data of the actigraphy.

**Table 3 ijms-24-04835-t003:** Stepwise regression models (backwards elimination) of the association of circulating endothelial biomarkers with circadian rhythm and orthostatic test variables for ME/CFS and healthy controls separately.

	ET-1, pg/mL		VCAM-1, ng/mL		ICAM-1, ng/mL	
Variables	ME/CFS (n = 32)	HCs (n = 29)	ME/CFS (n = 32)	HCs (n = 29)	ME/CFS (n = 32)	HCs (n = 29)
	B ^a^	B ^a^	B ^a^	B ^a^	B ^a^	B ^a^
Temperature						
Step 1 (initial model)						
Amplitude, °C	0.063	0.553	−60.484	44.436	−27.118	102.581
T_ IV, a.u.	−33.770	15.620	−1171.484	−8496.904	40.607	−18,658.229
T_R, a.u.	−0.552	0.416	542.522	138.721	66.280	429.192
T_PV, %	−0.034	−0.024	−2.726	−5.450	0.571	−7.151
T_M5, °C	0.537	0.013	−17.247	−0.443	2.547	−80.401
T_L10, °C	0.040	0.035	−49.908	8.894	−16.097	47.057
Final model						
T_ PV, %	**−0.034 ***	-	-	-	-	-
T_M5, °C	**0.564 ****	-	-	-	-	-
Amplitude, °C	-	**0.486 ***	-	-	-	-
Motor activity						
Step 1 (initial model)						
A_Mesor, a.u.	−0.002	0.002	0.107	−0.058	0.195	0.076
A_Amplitude, °C	0.000	0.002	−0.167	−0.203	0.205	−0.075
A_IV, a.u	0.524	−3.459	1082.393	136.465	171.076	170.145
A_R, a.u.	−0.287	−1.141	152.340	−352.068	22.964	203.005
A_PV, %	−0.054	−0.023	0.436	1.572	2.958	−1.292
A_M10, a.u.	0.001	−0.002	0.034	0.165	−0.245	−0.006
A_L5, a.u.	−0.001	0.002	0.564	−0.360	0.165	−0.119
Final model						
A_L10, a.u.	-	-	**0.665 ***	-	-	-
Passive standing test (NLT)						
Step 1 (initial model)						
Supine SBP, mmHg	−0.004	0.015	−2.117	−1.809	0.535	0.855
Supine DBP, mmHg	0.011	−0.033	−0.713	3.620	−0.352	7.752
Supine HR, bpm	0.014	−0.012	−2.286	5.209	1.605	5.633
ΔSBP_3F, mmHg	0.010	0.033	−4.509	6.352	−0.588	−0.913
ΔDBP_3F, mmHg	−0.069	−0.063	0.742	−2.246	−2.822	−5.600
ΔHR_3L, bpm	0.051	−0.011	−10.802	1.260	−2.959	2.758
ΔWT_3L, °C	−0.454	−0.053	−34.913	8.589	−11.224	26.378
Final model						
Supine SBP, mmHg	-	**0.022 ****	**−3.348 ***	-	-	-
Supine DBP, mmHg	-	**−0.033 ****	-	-	-	**5.290 ***
ΔSBP_3F, mmHg	-	**0.043 ***	-	-	-	-
ΔDBP_3F, mmHg	-	**−0.065 ****	-	-	-	-
ΔWT_3L, °C	**−0.374 ***	-	-	-	-	**31.939 ***

^a^ Values are unstandardized beta coefficients derived from stepwise regression models. Table shows Step 1 (initial model, which was conducted with all predictors entered together) and the predictors obtained in the final step after iteration (final model). Significance at * *p* < 0.05 and ** *p* < 0.01 is shown in bold. ET-1, endothelin-1; VCAM-1, vascular cell adhesion molecule-1; ICAM-1, intercellular adhesion molecule-1; ME/CFS, myalgic encephalomyelitis/chronic fatigue syndrome; HCs, healthy controls; NLT, 10-min NASA lean test; IV, intraday variability; R, Rayleigh test; PV, percentage of variance explained by the 24-h rhythm; M10 (or M5), maximum value of the variable; L5 (or L10), minimum value of the variable (see text for further explanation); SBP, systolic blood pressure; ΔSBP_3F, variation in SBP in the first three minutes of standing; DBP, diastolic blood pressure; ΔDBP_3F, variation in SBP in the first three minutes of standing; HR, heart rate; ΔHR_3L, variation in HR in the last three minutes of standing; ΔWT_3L, variation in wrist temperature in the last three minutes of standing. Variables starting with T_ or A_ were obtained, respectively, from the temperature or activity data of the actigraphy.

**Table 4 ijms-24-04835-t004:** Partial correlations between endothelial biomarkers and self-reported outcome measures for ME/CFS and matched healthy controls separately.

	ME/CFS (n = 32)						Healthy Controls (n = 29)					
Measures	ET-1, pg/mL		VCAM-1, ng/mL		ICAM-1, ng/mL		ET-1, pg/mL		VCAM-1, ng/mL		ICAM-1, ng/mL	
	*r*	*p*-Value	*r*	*p*-Value	*r*	*p*-Value	*r*	*p*-Value	*r*	*p*-Value	*r*	*p*-Value
FIS-40	0.529	**0.005**	−0.098	0.626	0.180	0.369	−0.165	0.430	−0.149	0.477	−0.322	0.117
HADS	0.407	**0.035**	−0.148	0.462	0.136	0.500	0.011	0.957	0.233	0.263	0.232	0.265
COMPASS-31	0.452	**0.018**	−0.104	0.607	0.042	0.834	0.063	0.764	0.046	0.826	−0.009	0.965
SF-36	−0.501	**0.008**	0.090	0.654	−0.205	0.304	0.140	0.503	−0.142	0.498	−0.052	0.805
OGS	−0.021	0.947	−0.166	0.587	0.178	0.561	0.127	0.529	−0.183	0.360	−0.049	0.810
PSQI	0.468	**0.014**	−0.296	0.134	−0.055	0.783	0.038	0.861	0.357	0.086	0.494	**0.014**

Values show correlation coefficients (r) and *p*-values from partial correlations adjusted for age and BMI. Significant *p*-values are shown in bold. ME/CFS, myalgic encephalomyelitis/chronic fatigue syndrome; ET-1, endothelin-1; ICAM-1, intercellular adhesion molecule-1; VCAM-1, vascular cell adhesion molecule-1; FIS-40, 40-item fatigue index scale; HADS, hospital anxiety and depression scale; COMPASS-31, 31-item abbreviated composite autonomic symptom score; SF-36, medical outcome study 36-item short-form health survey; OGS, orthostatic grading scale; PSQI, Pittsburgh sleep quality index.

**Table 5 ijms-24-04835-t005:** Stepwise regression models (backwards elimination) of the association of COMPASS-31 questionnaire with all the groups of studied variables for ME/CFS (n = 32) and healthy controls (n = 29) together.

	COMPASS-31	
	B ^a^	*p*-Value
Step 1 (initial model)		
Supine SBP, mmHg	−0.184	0.539
Supine DBP, mmHg	0.787	0.103
Supine HR, bpm	0.089	0.725
ΔSBP_3L, mmHg	−0.456	0.219
ΔDBP_3L, mmHg	0.527	0.338
ΔHR_3L, bpm	−0.320	0.287
ET-1, pg/mL	15.012	**0.000**
VCAM-1, ng/mL	0.064	**0.000**
T_Amplitude, °C	3.709	0.420
T_PV, %	0.610	0.062
T_M5; °C	−10.565	**0.009**
Age, years	−0.177	0.460
BMI, kg/m^2^	−0.132	0.804
Final model		
Supine DBP, mmHg	0.559	**0.028**
ET-1, pg/mL	14.727	**0.000**
VCAM-1, ng/mL	0.059	**0.000**
T_PV, %	0.657	**0.009**
T_M5, °C	−9.263	**0.007**

^a^ Values are unstandardized beta coefficients derived from stepwise regression models. Table shows step 1 (initial model, which was conducted with all predictors entered together) and the predictors obtained in the final step after iteration (final model). Significant *p*-values of the coefficients at the final model are shown in bold. ME/CFS, myalgic encephalomyelitis/chronic fatigue syndrome; BMI, body mass index; SBP_sp, systolic blood pressure in supine position; DBP, diastolic blood pressure; HR, heart rate; ΔSBP_3L, variation in SBP in the last three minutes of standing; ΔDBP_3L, variation in SBP in the last three minutes of standing; ΔHR_3L, variation in HR in the last three minutes of standing; ET-1, endothelin-1; VCAM-1, vascular cell adhesion molecule-1; T_Amplitude, amplitude of the temperature rhythm; T_PV, percentage of variance explained by the 24-h temperature rhythm; T_M5, nocturnal temperature (see text for further explanation). Variables starting with T_ or A_ were obtained, respectively, from the temperature or activity data of the actigraphy.

## Data Availability

The data presented in this study are available on request from the corresponding author.

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
