# Peer review of "Skin Temperature Circadian Rhythms and Dysautonomia in Myalgic Encephalomyelitis/Chronic Fatigue Syndrome: The Role of Endothelin-1 in the Vascular Tone Dysregulation"

_ijms, 2023, doi:10.3390/ijms24054835_

Round 1

Reviewer 1 Report

The manuscript by Cambras et al. investigates circadian rhythms and endothelial dysfunction in the context of autonomic dysfunction in ME/CFS.

Overall, I found the manuscript very well written and the methods employed appear to be appropriate to answer the question at hand. The study is well designed and given the statistical significance of many of the findings, it appears to be sufficiently powered. This is often difficult to achieve with such a heterogeneous disease. I found no issues with the science; however, there are a few typos and writing style issues that should be addressed prior to publication. There is a list of a few of them:

1. Through the manuscript, please check that only proper nouns are capitalized, unless the word starts a sentence. For example Chronic Fatigue Syndrome should not be capitalized.

2. Line 52 "Previous reports on autonomic dysfunction in ME/CFS suggest the imbalance" should be "an imbalance"

3. Line 83, This paragraph has a problem. I believe you either need to remove "since" at the beginning or remove the period after fer 19 and replace it with a comma.

4. Throughout the manuscript, make sure that when you use an abbreviation, it is articulated at the first time the word is used and then the abbreviation is used thereafter. ie. check triglycerides line 102 and 109

5. Line 105 you did not control the menstrual cycle, you controlled for it.

6. In the ledgend of the tables, you should either bracket or bold the letters that identify a section. ie Line 113, a Baseline self-reported outcome, should be a) Baseline etc.

7. Make sure that all counting numbers less than 10 are spelled out. ie. two minutes is wrong, but 10 minutes is correct.  Line 159

8. Make sure there is a space between a number and the degree symbol. ie. 28.5±0.37ºC should be 28.5±0.37 ºC.

Other than that it looks good!

Author Response

Response to reviewer #1 comments

The manuscript by Cambras et al. investigates circadian rhythms and endothelial dysfunction in the context of autonomic dysfunction in ME/CFS. Overall, I found the manuscript very well written and the methods employed appear to be appropriate to answer the question at hand. The study is well-designed and given the statistical significance of many of the findings, it appears to be sufficiently powered. This is often difficult to achieve with such a heterogeneous disease.

We want to thank the reviewer for all the helpful suggestions and we hope that this new updated version of the manuscript is now suitable for publication.

I found no issues with the science; however, there are a few typos- and writing style issues that should be addressed prior to publication. There is a list of a few of them:

  1. Through the manuscript, please check that only proper nouns are capitalized, unless the word starts a sentence. For example, Chronic Fatigue Syndrome should not be capitalized.

Thanks to the reviewer for this observation. It has now been corrected thoroughly manuscript.

  1. Line 52 "Previous reports on autonomic dysfunction in ME/CFS suggest the imbalance" should be "an imbalance"

Thank you for pointing out this grammatical error. It has been corrected appropriately.

  1. Line 83, This paragraph has a problem. I believe you either need to remove "since" at the beginning or remove the period after ref. 19 and replace it with a comma.

Thank you for this observation. It has now been corrected as you suggested.

  1. Throughout the manuscript, make sure that when you use an abbreviation, it is articulated the first time the word is used and then the abbreviation is used thereafter. i.e. check triglycerides line 102 and 109.

Thank you for this point raised by the reviewer. We have reviewed all abbreviations used throughout the manuscript.

  1. Line 105 you did not control the menstrual cycle, you controlled for it.

Unfortunately, we did not control the menstrual cycle among participants. Additionally, none of the participants were taking contraceptive pills, however; healthy volunteers were younger than ME/CFS patients, with 37.5% older than 50 years (menopause: 15/48= 31%), while in the ME/CFS cohort, the percentage was 50% (menopause: 23/67= 34%). As mentioned in the discussion, mean estrogen levels were higher in ME/CFS patients, although the mean age was higher in this group (also reported by Craddock TJA et al. PLOS One 2014). Further research studies are required to assess the role of sex hormone imbalance and autonomic dysfunction in ME/CFS. We have appropriately addressed a detailed rationale for this issue in the manuscript.

  1. In the legend of the tables, you should either bracket or bold the letters that identify a section. i.e., Line 113, a Baseline self-reported outcome, should be a) Baseline, etc.

It has been reviewed accordingly.

  1. Make sure that all counting numbers less than 10 are spelled out. i.e. two minutes is wrong, but 10 minutes is correct. Line 159.

Thanks for this observation from the reviewer. This has been now corrected in the manuscript.

  1. Make sure there is a space between a number and the degree symbol. i.e. 28.5±0.37ºC should be 28.5±0.37 ºC.

Thank you for this point. It has been corrected.

Reviewer 2 Report

For this reviewer, the manuscript is original and well organized. The introduction provides a thorough review and update on ME/CFS and clearly expresses what the aim of the study is.

The experimental design is well done, statistical analysis is adequate, experimental reasoning and data interpretation are sound. Finally, as this study aport evidence relating indicators of autonomic dysfunction through variables related with the vasoconstriction/vasodilation process and the subjective symptomatology of ME/CFS patients, associated with endotelial biomarkers, which may provide potential and novel therapeutic targets for this disesase, I recommend its publication.

As a comment for the authors, I suggest a better arrangement of the tables to avoid confusion in their reading and interpretation.

Author Response

Response to reviewer #2 comments

The manuscript is original and well-organized. The introduction provides a thorough review and update on ME/CFS and clearly expresses the aim of the study is. The experimental design is well done, statistical analysis is adequate, and experimental reasoning and data interpretation are sound. Finally, this study provides evidence relating to indicators of autonomic dysfunction through variables related to the vasoconstriction/vasodilation process and the subjective symptomatology of ME/CFS patients, associated with endothelial biomarkers, which may provide potential and novel therapeutic targets for this disease. I recommend its publication.

  1. As a comment for the authors, I suggest a better arrangement of the tables to avoid confusion in their reading and interpretation.

In agreement with the reviewer, we have arranged the tables for the readers as suggested.